# The YAP/TAZ Pathway in Osteogenesis and Bone Sarcoma Pathogenesis

**DOI:** 10.3390/cells9040972

**Published:** 2020-04-15

**Authors:** Heinrich Kovar, Lisa Bierbaumer, Branka Radic-Sarikas

**Affiliations:** 1St. Anna Children’s Cancer Research Institute, 1090 Vienna, Austria; lisa.bierbaumer@ccri.at (L.B.); branka.radic-sarikas@ccri.at (B.R.-S.); 2Department of Pediatrics, Medical University Vienna, 1090 Vienna, Austria

**Keywords:** Hippo pathway, mechanosignaling, bone development, osteosarcoma, chondrosarcoma, Ewing sarcoma, EMT, YAP, WWTR1, TAZ, EWS-FLI1, beta-catenin, microRNAs, verteporfin

## Abstract

YAP and TAZ are intracellular messengers communicating multiple interacting extracellular biophysical and biochemical cues to the transcription apparatus in the nucleus and back to the cell/tissue microenvironment interface through the regulation of cytoskeletal and extracellular matrix components. Their activity is negatively and positively controlled by multiple phosphorylation events. Phenotypically, they serve an important role in cellular plasticity and lineage determination during development. As they regulate self-renewal, proliferation, migration, invasion and differentiation of stem cells, perturbed expression of YAP/TAZ signaling components play important roles in tumorigenesis and metastasis. Despite their high structural similarity, YAP and TAZ are functionally not identical and may play distinct cell type and differentiation stage-specific roles mediated by a diversity of downstream effectors and upstream regulatory molecules. However, YAP and TAZ are frequently looked at as functionally redundant and are not sufficiently discriminated in the scientific literature. As the extracellular matrix composition and mechanosignaling are of particular relevance in bone formation during embryogenesis, post-natal bone elongation and bone regeneration, YAP/TAZ are believed to have critical functions in these processes. Depending on the differentiation stage of mesenchymal stem cells during endochondral bone development, YAP and TAZ serve distinct roles, which are also reflected in bone tumors arising from the mesenchymal lineage at different developmental stages. Efforts to clinically translate the wealth of available knowledge of the pathway for cancer diagnostic and therapeutic purposes focus mainly on YAP and TAZ expression and their role as transcriptional co-activators of TEAD transcription factors but rarely consider the expression and activity of pathway modulatory components and other transcriptional partners of YAP and TAZ. As there is a growing body of evidence for YAP and TAZ as potential therapeutic targets in several cancers, we here interrogate the applicability of this concept to bone tumors. To this end, this review aims to summarize our current knowledge of YAP and TAZ in cell plasticity, normal bone development and bone cancer.

## 1. Introduction

Invasion and metastasis are among the hallmarks of cancer [1]. However, although tumorigenesis is driven by a patient-specific diversity of de-novo mutations, the metastatic process converges on common cellular plasticity mechanisms of normal development hijacked by the cancer cell and driven by interaction with the cancer cell´s microenvironment. It is thus not surprising that many key molecules of metastatic progression are themselves rarely mutated but activated by intrinsic and extrinsic signaling cues in response to perturbed tissue homeostasis. Among these molecules are the transcriptional co-activators Yes-associated protein 1 (YAP-1, YAP) and its paralogue, the transcriptional co-activator with PDZ-binding motif (TAZ, WWTR1), which are typically controlled on the post-translational level by upstream regulatory signaling pathways in organ development and tissue homeostasis [2,3]. YAP and TAZ were reported to affect self-renewal and lineage commitment of stem cells [4,5,6,7], while hyperactivation of YAP and TAZ have been linked to cancer growth and metastasis in various tumors [8,9,10]. TAZ levels are elevated in approximately 20% of cancers and drive invasion and metastasis [11,12], while YAP is frequently overexpressed or amplified in cancer and was shown to promote resistance to chemotherapy in oncogene-addicted tumors upon inactivation of the oncogenic driver [13,14]. YAP and TAZ act as the ultimate nuclear effectors of signals transmitted from intrinsic biochemical and extrinsic biomechanical cues resulting from changes in the microenvironment and the architecture of tissues, including extracellular matrix rigidity, cell-cell and cell-matrix adhesion, cell density, shape and polarity, all of which play a role in metastatic dissemination and the homing of tumor cells to distant sites. Most recently, it was demonstrated that mechanosignal-transduction from stiffening surrounding extracellular matrix through YAP/TAZ is required to allow for oncogene-induced neoplastic reprograming of cells [15]. During the metastatic process, the highly plastic tumor cell shape is repeatedly changing between more round and compact and a more spread-out geometry. Cancer cells and their corresponding stroma co-evolve during disease progression, and increasing stiffening of the extracellular matrix assists metastatic spread. This transition is accompanied by intracellular tension and, as a consequence, translocation of YAP and TAZ from the cytoplasm to the nucleus. Nuclear accumulation is considered a sign of YAP/TAZ transcriptional activity, resulting in cellular reprograming [16,17]. Among typical YAP/TAZ targets are several secretory factors, including amphiregulin, connective tissue growth factor (CTGF) and cysteine-rich angiogenic inducer 61 (CYR61), which are important in stromal interactions and niche formation of cancer metastasis [18,19,20,21,22].

There is ample and rapidly growing literature on the mechanisms of YAP/TAZ regulation, particularly by the Hippo signaling pathway, which prohibits organ overgrowth through the regulation of cell proliferation and cell survival during normal development. Recent excellent and comprehensive reviews summarize this wealth of published information from different angles: from a protein structural perspective [23,24]; from the angle of tissue mechanics, microenvironment and cytoskeleton interactions [25,26,27]; from the aspect of DNA-binding protein partners of YAP/TAZ [28,29] and in the context of normal development [30]; specific cancers (i.e., of the breast [31], lung [32] liver [33], prostate [34], pancreas [35] and various pediatric cancers [36]) and other chronic and neurodegenerative diseases [37,38]. 

The adaptation to microenvironments of different tissue architecture plays a particularly important role in bone cancers which tend to metastasize through the hematogenous route and home primarily to bone and lungs, which are organs of completely different stiffness, extracellular matrix composition and oxygen supply. It is therefore not unexpected that the YAP/TAZ signaling pathway is prominently involved in bone cancer pathogenesis and metastatic spread. As this aspect has been poorly addressed so far, we will here first highlight key aspects of the growing complexity of YAP/TAZ signaling and ensuing phenotypes and then address their impact on normal bone development, bone cancer and metastasis, with a focus on chondrosarcoma, osteosarcoma, and Ewing sarcoma. 

## 2. Key Elements of YAP/TAZ Signaling

YAP and TAZ transmit a variety of upstream mechanical, architectural and metabolic signals from the plasma membrane to the nucleus. Initially, RAS homology family member A (RHOA) was identified as the critical regulator of YAP; however, among the RHO GTPases Rac family, small GTPase 1 (RAC1) and cell division cycle 42 (CDC42) were also found to be involved in its regulation [17,39,40]. In addition, there is crosstalk with various other signaling pathways, including NOTCH receptor (NOTCH), wingless (WNT) and bone morphogenetic protein (BMP) [41]. YAP and TAZ subcellular localization, stability and activity are regulated by protein phosphorylation, negatively on serines 127 and 397 (YAP) and 89 and 311 (TAZ) and positively on tyrosine 357 (YAP) and 316 (TAZ) [42]. Thus, YAP and TAZ are mainly regulated on the post-translational level (Figure 1), but little is known about the transcriptional regulation of their expression.

### 2.1. Transcriptional and Post-Transcriptional Regulators of YAP and TAZ

YAP is relatively uniformly expressed in most normal tissues except lymphoblastoid and myeloid cell types, whereas TAZ expression varies, with the lowest expression in blood. The TAZ promoter region contains binding sites for E26 (ETS), forkhead box (FOXO) and serum response factor (SRF) transcription factors and has been previously reported bound by ETS transcription factor variants ETV1, 4, and 5 in prostate cancer [43]. It is therefore possible, though not yet investigated, that aberrantly expressed ETS transcription factors in bone sarcomas (i.e., Ewing sarcoma) may affect TAZ transcription. TGFβ signaling was shown to selectively upregulate TAZ transcription through p38/MAPK-mediated activation of the transcriptional SRF co-activator myocardin-related transcription factor (MRTF) [44]. Finally, YAP in complex with β-catenin and the transcription factor TBX5 was demonstrated to regulate TAZ transcription [45]. For YAP, the only upstream regulatory transcription factor reported so far is cyclic adenosine monophosphate response element-binding protein (CREB) [46]. In cancer, both YAP and TAZ RNA are frequently increased, but the mechanisms behind this upregulation remain largely elusive and may involve both transcriptional and post-transcriptional mechanisms through suppression of the YAP/TAZ regulatory microRNAs, including miR-15a, miR-141-3p, mir-194, miR-195, miR-375, miR-381, miR-584, miR-125, miR-185, miR-9-3p and miR-129-5p, to name a few [47,48,49,50,51,52,53,54,55,56,57]. 

### 2.2. Serine and Tyrosine Phosphorylations Regulate YAP and TAZ Activity

The core Hippo pathway is initiated by activation of the serine/threonine protein kinases 4 and 3 (STK4/MST1 and STK3/MST2), the mammalian homologues of fruit fly Hippo, in complex with the MST1/2-activating scaffolding proteins Salvador family WW domain-containing protein 1 (SAV1) or Ras association domain family member (RASSF) at the plasma membrane [58]. MST1/2 phosphorylate a number of proteins involved in chromatin condensation, apoptosis and proliferation regulation, including the cytoplasmic large tumor suppressor kinases 1 and 2 (LATS1 and LATS2), in complex with the scaffolding protein MOB kinase activator 1 (MOB1). Activated LATS1/2 in turn bind and serine-phosphorylate YAP/TAZ, leading to their cytoplasmic retention through 14-3-3 and subsequent glycogen synthase kinase 3 beta (GSK3β) complex-mediated proteasomal degradation. Downstream of activated RAS signaling, however, RASSF scaffolding proteins divert LATS1/2 away from YAP/TAZ phosphorylation by complex formation with MST1/2 [59]. In addition, there is MST1/2-dependent and independent regulation of LATS1/2 and of YAP/TAZ phosphorylation in response to signals from tight and adherens junctions: WW domain-containing protein 1 (WWC1/KIBRA) and neurofibromin 2 (NF2/MERLIN) promote, while thyroid hormone receptor interactor 6 (TRIP6) inhibits, LATS1/2 phosphorylation through competition with MOB1 binding [60,61,62]. β1 integrin-dependent local activation of the small GTPase RAC1 at the plasma membrane to control the activity of P21 (RAC1)-activated kinase (PAK1) leads to NF2 phosphorylation, reducing its interactions with both LATS1/2 and YAP and ultimately promoting YAP nuclear translocation [63]. On the other hand, α/β integrin-mediated activation of focal adhesion kinase (FAK),which is essential for the establishment and turnover of integrin-matrix-based cell adhesions, may activate the nonreceptor tyrosine kinase SRC and the small GTPase CDC42, resulting in protein phosphatase PP1A-dependent dephosphorylation of YAP/TAZ [64]. In a feed-forward loop, YAP/TEAD activate transcription of thrombospondin (THBS), which acts upstream of FAK-activating phosphorylation [65]. Interestingly, computer-assisted modeling of mechanical networks controlling YAP suggested that the level of FAK activation might mediate the effect of substrate rigidity on YAP activation [66].

Moreover, YAP signaling is controlled by metabolic stress through the central energy stress sensor AMP-activated protein kinase (AMPK), which phosphorylates LATS1/LATS2, YAP on serine 94 and AMOTL to suppress YAP nuclear translocation [67,68,69]. 

In absence of serine phosphorylation, YAP and TAZ are stabilized, translocate to the nucleus and, upon activation by tyrosine kinases YES, SRC, LCK and likely others, co-activate TEA domain family (TEAD) transcription factors, which regulate a large number of genes encoding growth-promoting, apoptosis inhibitory proteins and structural and regulatory cytoskeleton components [45,70,71]. Specifically, SRC, downstream of FAK, appears to be an important YAP/TAZ regulatory signaling node, as it has been reported that, in addition to direct YAP/TAZ-activating tyrosine phosphorylation, it may repress LATS-mediated YAP/TAZ inhibitory serine phosphorylation by direct and indirect mechanisms in response to integrin-mediated adhesion to the extracellular matrix [70]. In addition to TEADs, YAP/TAZ bind a growing list of other transcription regulatory factors, including SMAD family member 3 (SMAD3), core-binding factor subunit beta (CBFB), tumor protein p73, Erb-B2 receptor tyrosine kinase 4 (ERBB4), early-growth response 1 (EGR-1), T-box transcription factor 5 (TBX5), hypoxia inducible factor 1 alpha (HIF1α) and RUNX family transcription factor 2 and 3 (RUNX2 and RUNX3) to regulate various aspects of growth and differentiation (for review, [25,28] and [72]). Importantly, there is also a growing list of evidence for YAP and TAZ interacting with chromatin-modifying and remodeling proteins, including switch/sucrose nonfermentable (SWI/SNF), Mediator, nucleosome-remodeling deacetylase (NuRD) and histone methylation complexes, as well as the GAGA factor, to regulate genome-wide target accessibility ([73] and, for recent review, [74]).

### 2.3. YAP/TAZ Cross-Talk with the WNT–APC–AXIN Pathway 

Canonical Wnt/β-catenin signaling plays a key role in bone remodeling, and perturbations of this pathway have been reported in bone cancer with aberrant activation observed in metastatic osteosarcoma and variable suppression observed in Ewing sarcoma (for a recent review, [75]). Beyond their role as transcriptional co-activators, YAP/TAZ are involved in Wnt/β-catenin signaling at various levels. At the cell membrane, they bind to Axin tethered to Wnt receptors frizzled/low-density lipoprotein receptor related protein 6 (LRP6), enabling recruitment of the beta-transducin repeat containing E3 ubiquitin protein ligase (BTRC) to the β-catenin destruction complex. Competitive replacement of YAP/TAZ from Axin by the binding of Wnts inactivates the destruction complex and leads to stabilization and nuclear translocation of both β-catenin and YAP/TAZ [76]. On the other hand, YAP/TAZ bind to the cytoplasmic Wnt signal transducer disheveled segment polarity protein 1 (DVL1) in the cytoplasm, inhibiting its phosphorylation and, consequently, abrogating its translocation to the nucleus [77,78]. Finally, serine-phosphorylated YAP and TAZ were reported to directly bind and retain β-catenin in the cytoplasm upon Hippo pathway activation [79,80]. On the other hand, tyrosine 357-phosphorylated YAP binds and activates a complex between β-catenin and the transcription factor TBX5 in the nucleus of β-catenin driven cancers to activate a number of anti-apoptotic genes, including *WWTR1* [45].

### 2.4. Role of MicroRNAs in YAP/TAZ-Regulated Circuitries

Several microRNAs are involved in the control of YAP/TAZ activity. For example, miR-135b-5p was demonstrated to promote osteogenic differentiation of mesenchymal stem cells (MSC) through targeting of LATS1 and MOB1, thus supporting YAP/TAZ nuclear translocation [47]. Additionally, miR-33-5p and -3p were implicated in osteogenic priming of MSC through indirect control of YAP and TAZ expression [81]. MicroRNAs miR-214-3p and miR-23a-5p shed from osteoclasts in exosomes downregulate osteoblast function by targeting upstream YAP regulatory fibroblast growth factor receptor and the YAP/RUNX2 transcriptional complex, respectively [82,83,84]. A circular RNA expressed from the second exon of the FAT Atypical Cadherin 1 gene (circFAT1) was demonstrated to sponge the YAP suppressive microRNA miR-375, thus upregulating YAP1 in osteosarcoma [52]. Vice versa, nuclear YAP/TAZ have also been reported to regulate microRNA biogenesis. For example, a directly YAP/TEAD-activated microRNA, miR-130, was found to target the competitive TEAD-binding protein vestigial-like family member 4 (VGLL4), thus amplifying YAP/TEAD target gene expression [85]. In human keratinocytes, nuclear YAP was demonstrated to sequester the microprocessor component DEAD-box helicase 17 (DDX17), thus downregulating microRNA processing at low cell density, while DDX17 was released, leading to increased microRNA processing when YAP was sequestered in the cytoplasm under conditions of high cell-cell contact [86], consistent with an earlier report suggesting increased mature microRNA biogenesis at high cell density [87]. In contrast, in a study of mammary epithelial MCF10A cells that did not discriminate between YAP and TAZ, their nuclear localization at low cell density was associated with repression of the negative DICER regulatory let-7 microRNA and a general activation of microRNA processing [88]. In how far tissue architecture and hippo signaling may contribute to the widespread downregulation of miRNA expression associated with human cancer remains to be established.

### 2.5. YAP and TAZ as Relays of Mechanosensors

YAP and TAZ are the principal triggers of numerous cell-autonomous responses but also implicated in mechanosensing and mechanotransduction, thus orchestrating interactions between tumor cells and the tumor microenvironment [15,27]. They capture information from the physical environment experienced by the cell and convert it into a transcriptional response. Mechanoregulation of YAP/TAZ depends on the structural organization and tension of the F-actin cytoskeleton, which receives stimuli through integrins, focal adhesions and other proteins indirectly involved in mechanosensation, including the cell polarity protein CRUMBS and the cell adhesion protein E-cadherin [27,89,90,91]. The ECM and cytoskeletal tension also profoundly impact on autophagic flux by YAP/TAZ directly promoting the expression of the GTPase-activating protein Armus. In normal and in tumor cells, YAP/TAZ-mediated autophagy in response to physical cues is responsible for dedifferentiation and the acquisition of self-renewing properties [92].

The ECM becomes stiffer during tumor progression, as well as in inflammatory and tissue damage processes, which trigger YAP/TAZ mechanotransduction; in response to mechanical tension, YAP translocates to the nucleus and gets activated. Thus, YAP/TAZ are induced by supra-physiological ECM rigidity [93]. Vice versa, it has been shown that the physiological “softness” of the ECM, as well as the cell compaction, inhibit YAP and TAZ [94]. Recently, it has been shown that microenvironments approximating the normal softness of healthy tissues prevent oncogene-mediated cell reprogramming and tumor emergence, events relying on YAP/TAZ mechanotransduction. Conversely, even subtle supra-physiological extracellular-matrix rigidity is transforming cells into tumor-initiating cells [15]. All these regulations rely on YAP/TAZ mechanotransduction, and YAP/TAZ target genes account for a large fraction of the transcriptional responses downstream of oncogenic signaling. 

YAP and TAZ are increasingly considered a precious readout to study mechanosensing and gain insights into signal exchanges between a tumor and its microenvironment [27,95], being the universal readers of mechanical inputs that cells receive from the surroundings. Interestingly, in organoid research, it has been shown that YAP is crucial for self-renewal and intestinal stem cell organoid formation. Colonies in stiff matrices featured nuclear YAP, unlike colonies grown in soft matrices [96,97], thus promoting organoid expansion. However, once a certain confluence is reached, the resulting compression leads to YAP inactivation and the impediment of further growth. All these observed particularities in mechanosensing profiles of YAP and TAZ promise a great opportunity for using mechanically controllable models to mimic physiological conditions. However, these sensitivities also raise concerns when it comes to the interpretation of the current wealth of data in the field of YAP/TAZ, which is largely obtained in mechanically uninformed experimental settings. 

### 2.6. YAP and TAZ—Functionally Redundant?

The body of evidence and number of regulatory pathways converging on YAP/TAZ and of YAP/TAZ effectors is constantly increasing. YAP and TAZ are frequently overexpressed in many cancers, and there is a growing interest in YAP and TAZ as potential prognostic factors and therapeutic targets. Therefore, many studies interrogate YAP/TAZ immunohistochemically for nuclear expression as a surrogate for activity. However, most antibodies used in the literature neither discriminate between the two paralogous proteins nor between the tyrosine phosphorylated active and the unphosphorylated inactive nuclear isoforms. Moreover, since both proteins are mostly co-expressed at variable proportions and have been demonstrated to bind and co-activate TEAD transcription factors, TEAD responsive luciferase reporter activity and the activation of TEAD target genes (i.e., CYR61 and CTGF) are frequently used as sole functional read-outs to demonstrate YAP/TAZ activity experimentally. 

However, there is good reason to believe that YAP and TAZ are not just functionally redundant proteins whose sole oncogenic role is restricted to TEAD activation. Knockdown of YAP and TAZ in hepatocarcinoma cells revealed common but also exclusive target gene spectra for the two distinct transcriptional co-activators [98]. Additionally, TEAD transcriptional activity is not solely dependent on co-activation by YAP/TAZ. For example, NF2/Merlin-orchestrated palmitoylation of TEAD independent of YAP/TAZ is required to stabilize and activate TEAD in response to cell contact [99]. In addition to TEAD, there are multiple and frequently co-expressed transcription factors that can partner with YAP and/or TAZ [28], but little is known about the mechanisms and factors that regulate the target specificity of YAP/TAZ-driven co-activation.

YAP and TAZ share important modular domains, including an N-terminal-binding domain for TEAD transcription factors; one (TAZ) or two (YAP) WW domains interacting with multiple PPXY-motif-containing regulatory proteins, including LATS1,2, ERBB4 and angiomotin (AMOT) family members and a C-terminal transactivation domain, including a coiled-coil region, which differs in sequence between YAP and TAZ, followed by a PDZ-binding motif. These shared protein interaction domains serve as docking sites for multiple proteins regulating YAP/TAZ subcellular localization and activity. For example, downstream of G-protein-coupled receptor (GPCR) signaling, AMOT is phosphorylated by LATS and released from filamentous actin at adherens junctions. In the cytoplasm, phosphorylated AMOT binds as a scaffold to MST1/2, LATS1/2 and YAP/TAZ, promoting their phosphorylation and preventing YAP/TAZ nuclear translocation [100,101,102]. However, in some cancers, including renal, hepatic and ovarian cancer, there is also evidence for a YAP-activating role of AMOT in the nucleus [103,104], and a recent study identified the involvement of an AMOT-binding long noncoding RNA in this nuclear activity [105]. In addition, tight junction protein 2 (TJP2/ZO-2) promotes the nuclear translocation of YAP/TAZ through interaction with the PDZ domain [106,107]. Exclusive to YAP are an N-terminal proline-rich domain interacting with tight junction protein 1 (TJP1) [108] and a central SH3 domain. Although the binding partners of the YAP SH3 domain are largely elusive, it may be responsible for the exclusive interaction of YAP with nuclear tyrosine-phosphorylated parafibromin (CDC73), while TAZ selectively binds to unphosphorylated parafibromin in complex with β-catenin [109]. In addition to acting as a transcriptional co-activator, there is evidence that TAZ plays a role in protein degradation through interaction with the E3 ligase BTRC [110]. Recently, an important biophysical difference between YAP and TAZ transcriptional activation mechanisms was described. The coiled-coil domain of TAZ, but not of YAP, is able to induce phase separation by forming liquid-like biomolecular condensates that compartmentalize and concentrate its DNA-binding cofactor TEAD4, coactivators BRD4 and MED1 and the transcription elongation factor CDK9 on TAZ-specific target genes. This ability was blocked by LATS-mediated TAZ serine phosphorylation [111].

On the organismal level, Yap knockout mice display embryonic lethality at day E8.5 with defects in the yolk sac vasculogenesis, chorioallantoic attachment and embryonic axis elongation. In contrast, mice lacking Taz develop polycystic kidney disease and emphysema, suggesting that Yap and Taz have nonredundant unique roles in vivo [77,112,113,114]. On the other hand, during early development of the pre-implantation embryo, Yap and Taz appear to substitute for each other, and Yap/Taz double-knockout leads to death prior to the morula stage [115]. YAP and TAZ have also been reported to play distinct roles in wound healing. Upon transforming growth factor beta 1 (TGFβ1)-induced myofibroblast transformation of keratinocytes after injury, YAP and TAZ were reported to inversely regulate alpha-smooth muscle actin and CTGF expression through distinct interactions with SMADs and TEADs [116]. In myoblasts, both Yap and Taz promote proliferation, but only Taz enhances myogenic differentiation later on [117]. With incremental knowledge about distinct targets and functions of YAP and TAZ, there is an increasing need for novel diagnostic biomarkers discriminating between them. Antibodies specific to tyrosine-phosphorylated (active) YAP and TAZ isoforms and tools assessing YAP or TAZ-specific differential expressions of genes beyond TEAD targets will help better appreciate their individual roles in normal development and disease. Figure 2 summarizes the current knowledge about shared and exclusive biological features of YAP and TAZ.

## 3. YAP/TAZ in Epithelial-Mesenchymal Transition (EMT)

In epithelial cancers, metastatic spread is initiated when tumor cells lose their junctions; reorganize their cytoskeleton, cell polarity and cell shape and undergo a change in their signaling and gene expression programs to promote interactions with the extracellular matrix, leading to migration and invasion. This process is well-known as epithelial/mesenchymal transition (EMT) and is not restricted to epithelial cancers but integral to normal embryonal development for the formation of mesoderm and the neural tube and to wound healing and fibrosis in the adult organism (for review, [118]). In cancer, EMT imparts cancer stem cell (CSC) characteristics, including invasiveness and intrinsic resistance to anoikis and genotoxic stresses, as well as self-renewing, colony forming and tumor-initiating capacities. EMT is a reversible process, and cells can revert back to an epithelial state by mesenchymal-epithelial transition (MET), which appears to be essential for the colonization of tumor cells at distant sites, a key step in metastasis [119]. Cells undergoing partial EMT resulting in a hybrid epithelial/mesenchymal state can give rise to circulating clusters of tumor cells, and display enhanced stemness, and therapy resistance (for review, [120]). Cell autonomous triggers, as well as paracrine signals derived from stromal cells such as fibroblasts and immune cells, regulate the EMT status. Based on the activity of typical EMT pathways, EMT-like processes have also been proposed to exist in cancers arising from nonepithelial/mesenchymal tissues, including brain tumors, hematopoietic malignancies and sarcomas [121]. The most recent example of a sarcoma recognized as undergoing a transient and reversible EMT-like process is Ewing sarcoma (for review, [122]).

In EMT, a number of interconnected signal transduction pathways communicate to sense physical and biochemical microenvironmental cues and mediate cellular plasticity at the basis of metastatic spread. Among them are a large number of receptor tyrosine kinases (RTK)-signaling through phosphatidylinositol-4,5-bisphosphate 3-kinase (PI3K)/AKT, including insulin-like growth factor 1 receptor (IGF1R); platelet-derived growth factor receptor (PDGFR) and fibroblast growth factor receptor (FGFR); TGFβ/BMP-signaling through SMAD3/4; NOTCH and its transcriptional effector hairy/enhancer-of-split related 1 (HEY1), WNT and its effector β-catenin and RHO/RAC/ROCK-signaling elicited by a myriad of receptors with tyrosine kinase or serine/threonine kinase activity or from G-protein-coupled receptors or integrins [118]. All these pathways cross-talk with mechanosensitive YAP and TAZ. For example, tissue mechanics and nutritional cues are integrated via IGF1R and PI3K/AKT activation, which is required for YAP/TAZ nuclear localization [72,123,124]. Moreover, in a comprehensive gain-of-function screen for directly YAP/TAZ-activating RTKs, FGFR, RET and MER proto-oncogene tyrosine kinase (MERTK) were found to phosphorylate YAP and activate TEAD-driven transcription [125]. In response to TGFβ signaling, phosphorylated complexes of SMAD2/3-4 directly interact with TAZ, which is required for their nuclear retention and transcriptional activity [126], while high YAP was reported to suppress BMP-induced phosphorylation and activity of SMAD1, 5 and 8 [127]. YAP/TAZ regulate genes of delta and jagged NOTCH ligands and of the core NOTCH transcription factor RBPJ (recombination signal binding protein for immunoglobulin kappa J region) and also cooperate with NICD (NOTCH intracellular domain) by direct physical interaction or synergistic co-activation of shared target genes [128,129]. The YAP/TAZ link to WNT/β-catenin signaling occurs at many levels through physical interaction with Axin, DVL and β-catenin, as discussed above in more detail. In addition, noncanonical Wnt signaling through receptor tyrosine kinase-like orphan receptor 1/2 (ROR1/2) couples EMT to self-renewal, proliferation, migration and chemotaxis via the recruitment of various adaptor proteins that trigger the activation of downstream Rho/Rac1 GTPases, PI3K/AKT and YAP/TAZ [130]. In parallel, ROR1/2 upregulate BMI1, a stem cell factor demonstrated to stabilize nuclear YAP1 in Ewing sarcoma [131]. Finally, in response to loss of cell polarity, ECM stiffening, focal adhesions and adherens junctions, a number of activated RTKs and GPCRs signal through Rho GTPases and polymerization of filamentous actin to inhibit LATS activity, eventually leading to YAP nuclear translocation. Loss or reduced expression of components of the cell junction-localized cell polarity protein complexes, including the scaffold proteins Scribble (SCRIB) or CRUMBS or of the members of the discs large homolog family (DLG1 or 5), as is frequently observed in cancer, inhibits MST/LATS/YAP interaction, resulting in YAP nuclear translocation, TEAD and SMAD activation and, eventually, the induction of EMT and cancer cell stemness [126,132,133]. In turn, YAP regulates a number of genes involved in upstream RHO/RAC/ROCK signaling [134,135]. Finally, the master transcription factors orchestrating EMT, snail family transcriptional repressors 1 and 2 (SNAI1/SNAIL and SNAI2/SLUG) and zinc finger E-box binding homeobox 1 (ZEB1), are able to form binary complexes with either YAP or TAZ that, together, control YAP/TAZ transcriptional activity [136,137]. In fact, *Snai1*/*Snai2* co-deletion in the mouse resulted in a loss of expression of several YAP/TAZ-TEAD target genes, including *Ctgf,* ankyrin repeat domain 1 (*Ankrd1*), AXL receptor tyrosine kinase (*Axl*), Dickkopf 1 (*Dkk1*) and *Cyr61* [137]. On the other hand, YAP physical interaction switches ZEB1 from a transcriptional repressor to a co-activator, promoting stemness, proliferation and the metastatic ability of breast cancer cells [138,139]. Taken together, YAP and TAZ appear to act as a central signaling hub, and a complex upstream regulatory network integrating diverse biophysical and biochemical inputs fine-tunes the phosphorylation status, activity, and effector spectrum of YAP and TAZ during EMT/MET. 

## 4. YAP and TAZ in Osteogenesis

It is likely that the function of YAP/TAZ in the metastasis of bone cancers at least partially recapitulates their role during normal development of the respective tissues of the origin. Although the exact developmental histogenesis for most malignant bone cancers has still not fully been resolved, they are considered to derive from mesenchymal cells arrested at some point of the endochondral bone formation (Figure 3).

### 4.1. Bone Formation and Skeletogenesis

Endochondral bone formation is initiated by the condensation of mesoderm- and neuroectoderm-derived MSC that differentiate into pre-chondroblasts to form a cartilaginous model of the bone. This model can be divided into four zones formed by resting, proliferating, prehypertrophic and hypertrophic chondrocytes. Development of this model is tightly regulated by parathyroid hormone (PTH)-related protein (PTHrP) and Indian hedgehog (IHH) produced by the proliferating or hyperthrophic chondrocytes, respectively, and by fibroblast growth factors. The coordinated expression of those factors and their receptors regulates either elongation of the skeletal element by keeping chondrocytes proliferating or inducing hypertrophy. Hypertrophic chondrocytes produce factors for vascularization, mineralize and then die by apoptosis. After vascularization, MSC differentiating into osteoblasts and precursors of osteoclasts immigrate, resorb the zone of provisional calcification and produce first woven bone, which is subsequently replaced by lamellar bone. 

Skeletogenesis is initiated during fetal development and persists throughout adult life as either a remodeling process in response to homeostatic regulation or as a regenerative process in response to physical injury. The mechanism of chondrocyte generation differs between neonatal and postnatal ages. At the neonatal age, bone growth is based on the consumption of chondroprogenitors. Postnatally, after the formation of the mineralized epiphyses, chondroprogenitors in an epiphyseal stem-cell niche acquire a unique PTHrP-positive skeletal stem cell phenotype, self-renew and proliferate and generate chondrocytes, which can transdifferentiate into osteoblasts and stromal cells (for review, [140]). Removal of woven or old bone by resorption is regulated by the receptor activator of NF-κB ligand (RANKL), which binds to its receptor RANK to stimulate osteoclast differentiation and activity [141]. Thus, bone is subject to permanent remodeling, a process of continuous resorption and neosynthesis of bone, in which osteoblasts regulate osteoclast formation and function by expressing the negative regulator osteoprotegerin (OPG) or positive regulators RANKL and macrophage colony-stimulating factor (MCSF), determining bone structure and quality during adult life [142]. Conversely, the transition from bone resorption to bone formation during bone remodeling is mediated by less-understood osteoclast-derived coupling factors, which act on bone-forming cells and modulate their bone-forming activity.

### 4.2. YAP and TAZ as Regulators of Osteogenesis

Thus, at the basis of endochondral bone formation are pluripotent and skeletal MSC, which, depending on the mechanical and biochemical microenvironment, commit to either chondroblast or osteoblast differentiation, while adipogenic, neurogenic, myogenic and hematopoietic supportive stroma differentiation is suppressed. In the growth plates, YAP is expressed in resting and proliferative chondrocytes but less in hypertrophic chondrocytes. In contrast, the expression of Ser-phosphorylated, inactive YAP is strongest in the hypertrophic chondrocytes [143]. In addition to chemical [144,145] and metabolic [146] factors, physical cues direct osteogenesis from MSC [8,147]. It has been shown that nanoscale variations in extracellular matrix (ECM) compositions, the spacing of ECM nanodomains and matrix elasticity are sufficient to determine the differentiation fate of MSC with stiff matrices driving preferentially osteogenic differentiation [148,149]. It is therefore not surprising that YAP/TAZ play an important role in osteogenesis. Moreover, incorporation of α-smooth muscle actin in contractile stress fibers of MSC favor osteogenic differentiation in a YAP-dependent fashion [150]. In fact, YAP depletion was shown to downregulate the expression of osteogenic genes and of focal adhesion components paxillin, β-1 integrin and zyxin in human MSC in vitro [151]. Vice versa, β-1 integrin–dependent cell adhesion was demonstrated to be critical for MSC cell proliferation dependent on the local activation of the small GTPase RAC1 at the plasma membrane and phosphorylation of NF2/Merlin by p21-activated kinase 1 (PAK1), leading to its dissociation from the LATS1/YAP or TAZ complex (the study used an antibody not discriminating between the two paralogues), eventually resulting in the dephosphorylation and nuclear translocation of YAP and/or TAZ [63].

SNAIL and SLUG have recently been proposed to selectively modulate self-renewal, pluripotency and proliferative activity of embryonic and of several epithelial and mesenchymal tissue and cancer stem cells without affecting the expression of stem cell transcription factors NANOG, SRY-box transcription factor 2 (SOX2) and POU class 5 homeobox 1 (POU5F1/OCT4). However, a number of canonical YAP/TAZ-TEAD target genes are inhibited upon SNAI1/SNAI2 loss. As a consequence, *Snai1*/*Snai2* double-knockout in mice did not affect the osteoprogenitor lineage commitment of MSC but impaired their ability to complete the terminal steps of their osteoblastogenic program paralleled by the protein destabilization of Yap/Taz [136,137]. 

In mice, Yap promotes early growth plate chondrocyte proliferation by activating Sox6 expression and inhibits subsequent chondrocyte maturation by suppressing the expression of collagen type 10 (Col10) [127,143]. A recent study in which YAP and/or TAZ were inactivated at different times of mice embryonal development by knockout revealed opposing effects of YAP and TAZ at different stages of osteoblast development: in osteoblast progenitors, they block differentiation towards the osteoblast lineage by repressing RUNX2 and β-catenin activity, but, in mature osteoblasts and osteocytes, they promote bone formation and inhibit bone resorption [152]. In contrast, conditional knockout of YAP or TAZ in young adult mice demonstrated that YAP stabilizes nuclear β-catenin, while TAZ binds to SMAD4 co-activating RUNX2 to drive osteoblast differentiation of MSC and inhibit adipogenic peroxisome proliferator-activated receptor gamma (PPARγ)-induced gene transcription [153,154]. Interestingly, complexes between RUNX2, YAP/TAZ and SNAIL/SLUG were found associated with the promoter regions of multiple genes involved in osteoblastic differentiation and function in mice [137]. Additionally, in osteoblastic differentiation of human MSC, the importance of YAP has recently been established [155].

Bone remodeling in adult organisms relies on the balanced activity of osteoblasts and osteoclasts. RANKL expressed by osteoblasts, bone marrow stromal cells and activated T lymphocytes binds to RANK on the surface of osteoclast precursor cells or mature osteoclasts to promote osteoclast differentiation and bone resorption. In turn, activated osteoclasts release exosomes loaded with osteoblast-inhibitory microRNAs. These include miR-23a-5p, which targets the RUNX2/YAP transcriptional complex [82], and miR-214-3p, which targets, among others, the YAP/TAZ regulatory FGF receptor and osteoblastic transcription factor ATF4 [83,84]. Dysregulation of the RANKL/RANK pathway is involved in bone destructive diseases such as osteoporosis, rheumatoid arthritis and cancer bone metastasis. YAP/TAZ signaling is also involved in bone remodeling in response to the mechanical load, which is sensed by PIEZO1, a nonselective Ca^2+^-permeable cation channel expressed on osteoblasts, resulting in YAP nuclear translocation and the expression of YAP/RUNX2 target genes. Among them are type II and IX collagens, whose deposition in the extracellular matrix in turn regulate bone resorption by osteoclasts [156]. 

Besides their role in balancing the differentiation fate of MSC during developmental skeletogenesis and bone regeneration, YAP/TAZ are also essential for MSC migratory potential. In fact, migrating MSCs represent a source of multipotent cells that are available for the repair of damaged tissues and organs (for example, bone fractures) [157]. During early vertebrate development, the migration and ingression of mesoderm- and neural crest-derived MSC initiate limb bud development [158]. Although the molecular mechanisms are still poorly understood, the limb bud identity transcription factor TBX5, which has been shown to be co-activated by YAP/TAZ in a feed-forward loop with FGFR signaling [159,160], seems to play a pivotal role [161,162]. Furthermore, the mechanical growth factor (MGF), encoded by a splice variant of IGF1 and upregulated by mechanical stimuli in the epiphyseal growth plate, activates YAP in the nucleus through RhoA GTPase-mediated cytoskeleton reorganization to promote focal adhesion formation and the migration of chondrocytes [163].

Taken together, YAP/TAZ are key players in balancing pluripotency, self-renewal, proliferation, differentiation and migration/invasion during the normal osteogenesis of vertebrates. These same functions are involved in the EMT/MET regulation of bone cancers. Thus, important roles for YAP/TAZ in bone cancer biology, specifically metastasis, can be predicted. 

However, depending on the developmental timing of the derivation of a bone cancer from the mesenchymal lineage during embryonal or adult osteogenesis, these roles may vary. Chordoma localized along the skeletal axis in bones from the skull base to the coccyx is considered to derive from early developmental mesodermal cell types in notochord remnants [164]. Chondrosarcoma, the most common malignant bone tumor in adults, is considered to derive from benign enchondroma or osteochondroma, which are thought to arise from rests of growth plate cartilage or chondrocytes [165]. For Ewing sarcoma, amply available genomic and experimental data point to mesoderm- or neuroectoderm-derived MSC, presumably at an osteochondrogenic progenitor state, as the cell-of-origin [166,167,168,169,170,171,172]. For osteosarcoma, there is evidence for derivation from pre-osteoblasts or osteoblasts with the contribution of microenvironmental mesenchymal stem cells (MSC) (for review, [173,174]). Similarly, a giant cell tumor of bone, a rare, locally aggressive tumor which occasionally metastasizes to the lung, is of osteoblastic lineage [164]. 

## 5. YAP/TAZ in Chondrosarcoma

Chondrosarcoma is a cartilaginous neoplasm arising primarily in preexisting long bones and the pelvis in adults and accounts for 20% of all bone cancers. Nonconventional chondrosarcoma variants include clear cell chondrosarcoma, mesenchymal chondrosarcoma and dedifferentiated chondrosarcoma. Normal chondrocyte differentiation is regulated by YAP at multiple steps. It promotes early chondrocyte proliferation through the TEAD-mediated activation of SOX6 but inhibits subsequent chondrocyte maturation through the interaction with RUNX2 and blocking SOX9 and Col10a1 [143,175]. It is therefore likely that the dysregulation of YAP/TAZ may contribute to the pathogenesis of chondrosarcoma. However, except for some indirect evidence of a possible involvement of YAP/TAZ in chondrosarcoma aggressiveness, little is known about YAP/TAZ in this disease. For example, pigment epithelium-derived factor (PEDF), which is frequently low in tumors and whose overexpression reduced orthotopic growth and metastasis in osteosarcoma [176], induced a decrease in p-AKT, p-ERK, p-FAK, RHOA and CDC42 protein levels in chondrosarcoma cells associated with decreased adhesion and invasion [177]. Further, high expression levels of the gamma isozyme of protein phosphatase 1 have been reported in chondrosarcoma, which by inference may contribute to YAP/TAZ activation through dephosphorylation [178]. In addition, inhibition of the YAP/TAZ-activating kinases SRC and RAC was reported to interfere with the migratory and invasive properties of chondrosarcoma cells [179], while BMP-7-induced SRC activation promoted chondrosarcoma cell migration [180]. Although YAP/TAZ were not investigated in these studies, it may be speculated that the observed molecular alterations affected chondrosarcoma adherence and invasion phenotypes through abrogated YAP/TAZ signaling. A recent study identified YAP nuclear accumulation as a consequence of LATS1 inactivation by protein arginine methyltransferase 1 (PRMT1) as an independent bad prognostic factor in chondrosarcoma [181]. Finally, the treatment of chondrosarcoma cells with the BET-bromodomain inhibitor JQ1 downregulated YAP/TAZ and LATS1, leading to the upregulation of cyclin-dependent kinase inhibitor 1a (CDKN1A/p21) and cell cycle arrest, senescence and apoptosis [182]. Potentially, this downregulation was mediated by epigenetic restoration of the expression of the inhibitory YAP/TAZ-binding scaffold protein angiomotin (AMOT) [183].

## 6. YAP/TAZ in Osteosarcoma

Osteosarcoma generally affects the metaphyses of long bones. The majority of cases arise in children and adolescents younger than 20 years of age. Aggressive high-grade tumors account for approximately 90% of osteosarcomas. Despite improvements in adjuvant and neoadjuvant therapy in conjunction with surgery, less than 30% of patients with metastasis can be cured. 

Immunohistochemically, high YAP expression was noted in 60% to 85% of osteosarcomas [184,185,186,187], but only nuclear immunoreactivity in 30% to 46% of cases correlated with an adverse outcome (observed in some [184,185] but not all studies, potentially due to small patient cohorts [186,188]). Metastatic events were either not [186] or only slightly more frequent in patients with nuclear YAP (53%) than in those lacking nuclear YAP immunoreactivity (36%) [185]. However, knockdown of YAP in osteosarcoma cell lines suppressed in vitro tumor cell proliferation, clonogenicity and invasion, as well as tumor formation in mice [186,188]. Likewise, the inhibition of upstream RHO/ROCK signaling was demonstrated to reduce nuclear YAP protein and TEAD target gene expression and abolish the in vitro clonogenic and migratory and in vivo pulmonary metastatic ability of osteosarcoma cell lines in immunodefficient mice. Consistent with its important role in osteosarcoma pathogenesis, the modeling of osteosarcoma in p53 heterozygous mice by activating hedgehog signaling in mature osteoblasts revealed the activation of Yap, and Yap knockdown significantly reduced the tumorigenic potential of osteosarcoma cells in this model [187]. Of note, several hedgehog-signaling components, including smoothened (SMO), GLI family zinc finger proteins 1 and 2 (GLI1 and GLI2) and patched 1 (PTCH1), were found overexpressed in human osteosarcoma, and knockdown of GLI2 recapitulated the YAP knockdown effect in suppressing the proliferation, invasion and EMT of osteosarcoma cell lines [189,190]. Similarly, inactivation of the Lats1/2-activating adaptor proteins Mob1a/1b or the heterozygous loss of Nf2 in a p53^+/−^ background resulted in osteosarcoma development with frequent metastases in about 24% and 63% of knockout mice, respectively [191,192,193]. Moreover, high expression levels of PP1 isoform gamma and a high rate of promoter methylation and/or the downregulation of members of the RASSF tumor suppressor family (RASSF4, 5 and 10), which negatively regulate YAP/TAZ nuclear translocation, were reported in osteosarcoma cell lines and tumors [178,194,195]. For RASSF5, low expression levels were correlated with the presence of distant metastases, and ectopically forcing the expression of any of them resulted in osteosarcoma cell apoptosis.

While all these data are consistent with an important role of activated nuclear YAP in osteosarcoma, there is little data on the potential role of TAZ in human patients. In a small series of 16 human osteosarcomas, 13 showed TAZ expression higher than normal bone tissue or chondromas [196]. A recent study performed in dogs suggested that TAZ may be specifically associated with the proliferative and migratory ability of metastasis-derived osteosarcoma cells [197]. Consistent with this finding, sublines of human osteosarcoma cell lines U2OS and HOS selected for gradually increasing migratory ability showed parallel increases in TAZ RNA and protein expression and of EMT markers N-cadherin, vimentin and SNAIL, while E-cadherin was gradually decreased. This phenotype was reversed by TAZ knockdown, while ectopically forcing TAZ expression promoted the EMT-like transition of osteosarcoma cell lines [196]. The study also provided experimental evidence that much of the phenotypically EMT promoting effect was mediated by the directly TAZ/TEAD-activated microRNA hsa-miR-135b predicted to target LATS2, APC and GSK3β 3´UTRs and thereby upregulating TAZ in a positive feedback loop. Hsa-miR-135b overexpression was demonstrated to increase osteosarcoma xenograft growth [196]. Likewise, hsa-miR-224 was described as a TAZ/TEAD directly-activated target gene that enhances the proliferation and tumorigenicity of osteosarcoma and was demonstrated to rescue the migration, invasion and tumorigenicity of osteosarcoma cells upon TAZ knockdown at least in part through targeting SMAD4 [150]. Thus, it is possible that YAP and TAZ serve distinct functions in osteosarcoma pathogenesis and progression.

## 7. YAP/TAZ in Ewing Sarcoma

Ewing sarcoma typically arises in the shaft of long bones but may also affect flat bones and involve soft tissue. Although the majority of patients present with localized disease and only about 20% of cases have clinically overt metastases at diagnosis, almost all patients relapse with distant metastases in the absence of chemotherapy, suggesting the early development of micro-metastatic disease in most patients. Phenotypically and genetically, Ewing sarcoma is a remarkably homogenous disease. It most likely derives from a single mesoderm- or neural crest-derived mesenchymal cell lineage with variable developmental timing of differentiation arrest and oncogenic initiation, as suggested by heterogeneity along a mesenchymal differentiation gradient, but lack of distinct subgroups in a bulk epigenome analysis of primary tumor samples [198]. The oncogenic driver of the disease is an Ewing sarcoma breakpoint region 1/E26 (*EWSR1-ETS*) gene fusion leading to the expression of a potent, aberrantly cell reprograming transcription factor, most frequently EWS-FLI1 (Ewing sarcoma RNA binding protein-Friend leukemia virus integration site 1). Various attempts to model Ewing sarcoma in mice failed due to toxicity of the fusion oncogene to most cell types [199]. Among the few cell types tolerant to neoplastic EWS-FLI1 transformation in the mouse are osteochondrogenic progenitors of the embryonal bone superficial zone [170]. In these cells, YAP and TAZ regulate the expression of secreted ECM proteins proteoglycan 4 (PRG4) and tenascin C (TNC) downstream of CDC42 signaling [200]. 

EWS-FLI1 activates and represses almost equal numbers of genes either directly by chromatin-binding at gene promoters and enhancers or indirectly via interaction with other transcriptional circuitries [201]. Unique to EWS-ETS transcription factors and dependent on the self-association properties of the intrinsically disordered EWS N-terminus is their ability to form phase-separated clusters on GGAA microsatellites, turning them into de-novo transcriptional enhancers [201]. 

While there is ample experimental evidence for high EWS-FLI1 levels activating cell cycle- and suppressing differentiation-associated genes, thus driving self-renewal and proliferation of Ewing sarcoma cells, recent studies indicated that low EWS-FLI1 levels may promote Ewing sarcoma migration, invasion and metastasis [122,202,203]. Among EWS-FLI1-modulated genes whose expression increases upon EWS-FLI1-low conditions are lysyl oxidase (LOX) [204,205], TNC [206] (although there is also data suggesting *TNC* as an EWS-FLI1-activated gene [207,208]) and the TGFβ receptor TGFBR2 [209,210]. A recent study used LOX as a surrogate marker of low EWS-FLI1 expression and identified such cells to be present at an incidence of 2% in primary Ewing sarcomas [122]. LOX was previously reported a biomarker of lung and liver metastasis in colon cancer, where it correlates with YAP expression [211]. In addition, it was shown to promote tissue stiffening, thus inducing a premetastatic niche and metastasis in breast cancer [212]. It is therefore possible, although still hypothetical, that LOX-mediated shaping of ECM stiffness may affect YAP/TAZ signaling and metastasis in Ewing sarcoma with low EWS-FLI1 expression as well. TNC is an extracellular matrix glycoprotein that promotes pathogenesis and the progression of many cancers through various signaling pathways, including YAP/TAZ. TNC is abundantly expressed in most Ewing sarcomas, and high expression levels correlate with adverse outcomes. Furthermore, TNC expression in Ewing sarcoma increases upon microenvironmental stress (hypoxia or nutrient deprivation) [213]. Interestingly, while in osteosarcoma, TNC binding to α9β1 integrin promoted metastasis by repressing YAP nuclear translocation and target gene activation [214], TNC binding to α5β1 integrin in Ewing sarcoma promoted metastasis by activating YAP [207], potentially through activating tyrosine phosphorylation by SRC kinase [213]. This discrepancy may reflect the derivation of these two types of bone sarcoma from the mesenchymal lineage at different times of development, as discussed above. 

Reactivation of TGFβ signaling in EWS-FLI1 low cells through the upregulation of TGFBR2 may contribute to Ewing sarcoma metastasis through a mechanism involving TAZ. It has been previously shown that TGFβ induces TAZ transcription but not YAP expression in both epithelial and mesenchymal cells through p38 MAPK-mediated phosphorylation and the activation of the serum response factor (SRF) transcriptional co-activator myocardin-related transcription factor (MRTF) [44]. Interestingly, we and others have seen a selective increase in TAZ but not YAP transcription upon the knockdown of EWS-FLI1 in Ewing sarcoma cells (Bierbaumer et al., in revision; [215]). Our recent study revealed that EWS-FLI1 occupies and blocks YAP/TEAD-bound sites on regulatory regions of genes involved in EMT-like cytoskeletal reprograming [203]. Additionally, these sites were found enriched in activator protein 1 (AP-1)-binding motifs, consistent with a previous report describing a genome-wide association of YAP/TAZ/TEAD and AP-1 at gene enhancers [216]. Although it remains unclear how EWS-FLI1 blocks YAP/TAZ/TEAD target gene activation, it is worth noting that we found EWS-FLI1 repressed chromatin regions enriched in AP-1-binding motifs and that EWS-FLI1 physically interacts with AP-1 [217,218]. Downstream of Rho-F-actin signaling the transcriptional co-activator MRTFB is enriched on YAP/TEAD-bound target genes upon the knockdown of EWS-FLI1, which get activated and drive the phenotypic switch from a proliferative to a highly migratory phenotype [203]. The study implied that EWS-FLI1 does not interfere with upstream GPCR and Rho GTPase signaling in Ewing sarcoma but at the level of transcriptional co-activation of TEAD. 

While fluctuations in EWS-FLI1 levels and the ensuing consequences for YAP/TAZ signaling may be the basis for the metastatic potential of a small fraction of tumor cells in Ewing sarcoma (2% according to the postulated biomarker LOX [122]), they may not fully explain Ewing sarcoma aggressiveness. First, while there is a recent single-cell RNA-sequencing-based demonstration of cell-to-cell heterogeneity in EWS-FLI1 RNA levels associated with distinct transcriptional programs [219], direct evidence for variations in EWS-FLI1 protein levels in Ewing sarcoma is still missing. Second, there is evidence for YAP being required for sustained Ewing sarcoma cell proliferation and anchorage-independent growth [131]. Third, YAP and TAZ, although variably expressed in Ewing sarcoma cell lines and tumors, are consistently found in the cytoplasm and the nucleus in bulk tumor cell populations, suggesting that the incidence of their nuclear expression in individual Ewing sarcomas is high. A similar result was obtained for bone marrow-derived MSC, suggesting that YAP/TAZ expression and activity patterns in Ewing sarcoma derive from its putative cell-of-origin. Consistent with this supposition, no genetic alterations were found in genes encoding for any of the core Hippo/YAP/TAZ-signaling components in Ewing sarcoma. However, high levels of the PP1 subunit PP1c [220], epigenetic promoter silencing of the upstream negative YAP regulatory proteins RASSF2 and RASSF1A [221,222] and, as a consequence, expression from a cryptic downstream promoter of an oncogenic RASSF1 variant, RASSF1C, containing an alternative exon 1 were observed in Ewing sarcoma. RASSF1C promotes SRC kinase-mediated tyrosine phosphorylation and the transcriptional activation of YAP and may therefore be responsible for persistent YAP nuclear activity in Ewing sarcoma [215]. Moreover, the polycomb ring-finger protein BMI-1 was demonstrated to be variably but consistently expressed and to stabilize nuclear YAP in Ewing sarcoma [131,223]. In how far this effect is mediated by the suppression of the RASSF1A promoter remains to be elucidated. 

In a series of Ewing sarcoma samples analyzed by immunohistochemistry with an antibody that does not discriminate between YAP and TAZ and is ignorant to their phosphorylation status, 40% of primary tumors and ~70% of relapsed tumors and metastasis samples showed high YAP/TAZ expression, significantly correlating with a shortened time to relapse and adverse overall survival [215]. These data suggest that, while fluctuations in EWS-FLI1 may be involved in the initiation of the metastatic process by inducing EMT/MET-like processes involving YAP/TAZ, persistent YAP/TAZ activity is selected during Ewing sarcoma metastasis. Yet, nuclear YAP and TAZ are only infrequently found in Ewing sarcoma immunohistochemistry (20% and 15%, respectively) [150]. 

Among GPCRs, the chemokine receptor CXCR4 has been linked to metastasis formation in many cancers, including Ewing sarcoma [224]. Here, CXCR4 was demonstrated to be heterogeneously expressed and upregulated upon stress, resulting in increased migration, which could be blocked by small molecule inhibitors to the Rho GTPases CDC42 and RAC [225]. Since RAC was shown to activate YAP in response to β-integrin signaling in MSC [63] and CXCR4 was reported to activate YAP in other cancers (i.e., non-small cell lung cancer [226]), it is possible that the metastatic spread of Ewing sarcoma cells with high CXCR4 may be supported by a mechanism involving YAP/TAZ.

In addition, the zyxin-related cell adherence junction protein TRIP6, which is known to activate YAP/TAZ through competition with MOB1 binding of LATS1/2 [62], is highly overexpressed in Ewing sarcoma and associated with a pro-invasive gene signature. Experimental suppression of TRIP6 expression impaired the migratory, clonogenic and tumorigenic potential of Ewing sarcoma cell lines [227].

## 8. Rare Bone Cancers

Nothing is known about the YAP/TAZ pathway in the giant cell tumor of bone, a locally aggressive and rarely metastasizing neoplasm which typically arises in skeletally mature young adults at the end of long bones. It is typically composed of multinucleated osteoclast-like giant cells, mononuclear macrophage-like osteoclast precursor cells and mononuclear osteoblast-derived spindle-shaped stromal cells characterized by histone H3.3 (H3F3A) mutations, which are considered the neoplastic tumor component [228]. Thus, it cannot be excluded that the ensuing general epigenetic deregulation may also affect YAP/TAZ pathway components.

Although not a bone but a soft-tissue sarcoma arising from endothelial or pre-endothelial cells, hemangioendothelioma is most frequently localized in bone, lung or liver and mentioned here for its model role as an unambiguously YAP/TAZ-driven neoplasm. Here, YAP or TAZ are activated by gene rearrangements with either the transcription factor TFE3 (YAP-TFE3) or the calmodulin-binding transcription activator CAMTA1 (TAZ-CAMTA1) and retaining the YAP/TAZ TEAD-binding and WW domains [229,230,231]. Although the fusion protein can still be phosphorylated by LATS, the presence of a nuclear localization domain in the fusion partner suffices to keep the protein in the nucleus and constitutively activate the YAP/TAZ transcriptome [232]. 

## 9. Therapeutic Opportunities

As YAP/TAZ activity has been associated with cancer stem cell renewal, tumor growth, invasion and metastasis in many cancers [8,9,10,11,12], there is hope that strategies targeting the YAP/TAZ regulatory network may lead to novel, efficient cancer therapies. A number of small molecule inhibitors or drugs have been discovered that interfere with YAP/TAZ signaling at various levels. However, with the exception of verteporfin discussed below, no direct inhibitors of YAP/TAZ have been reported so far. This may in part be due to the extremely large interaction surface of YAP/TAZ with TEAD [233,234], which makes it difficult for small molecules to break it [235]. Additionally, drug-screening approaches may have suffered from not sufficiently accounting for potential differences in the biological activities of YAP and TAZ in different cellular contexts, producing difficult-to-interpret results. For example, a large compound screen for TAZ inhibitors in osteosarcoma used the subcellular localization of GFP-tagged TAZ and TEAD-responsive reporter gene activity as readouts. Not surprisingly, there was little overlap between compounds shifting TAZ subcellular localization and those altering either TEAD reporter activity and/or phosphorylation status, as TEAD activity in this cell type may be selectively driven by YAP and not TAZ [236]. In fact, studies attempting to therapeutically break YAP/TAZ protein interactions so far focused on TEAD as binding partner. Using fragment-based and computational modeling approaches, the Ω-loop or the α-helix protein interaction domain on YAP, and a hydrophobic palmitoylation pocket on TEAD were identified as targetable domains [233,234,237]. Peptides binding to this pocket may serve as scaffolds for the development of compounds with the potential to break the YAP/TAZ-TEAD interaction [238], and several such compounds are currently in pharmaceutical development [239,240,241,242] (Figure 4). 

The porphyrin compound verteporfin is an FDA approved drug (Visudyne) for the photodynamic treatment of age-related neovascular macular degeneration. Here, nonthermal red light induces the formation of cytotoxic oxygen species, which leads to local endothelial cell damage [243]. Independent of this light-activated function, verteporfin was suggested to specifically bind to YAP (and, potentially, also TAZ), inducing conformational changes that promote cytoplasmic sequestration by 14-3-3σ, preventing their interaction with TEAD and subsequent target gene activation by a mechanism that does not require the generation of reactive oxygen radicals [14,244] (Figure 4). Promising effects on cell survival/proliferation and the migration in models of various cancers, including osteosarcoma, were reported with verteporfin treatments at concentrations in the micromolar range [185,245,246]. However, our own studies performed in Ewing sarcoma cell lines showed that IC50 values differ several-fold between experiments performed in standard, artificial light laboratory conditions and those performed in the dark and that strong antimigratory effects can be seen already at nanomolar concentrations well below IC50 values. Thus, it appears that concentration-dependent effects discriminate between in vitro pro-apoptotic and the antimigratory/invasive activity of verteporfin, potentially reflecting distinct requirements for YAP/TAZ-inhibition and ROS-generation of these distinct outcomes. Verteporfin has not been used for the systemic treatment of patients yet. A short half-life (5 to 6 h) and maximally achievable plasma concentrations of 6–20 mg/m^2^ may hamper its clinical application to cancer treatment. We found in an orthotopic xenotransplantation mouse model that a daily systemic administration of verteporfin at a dose of 25 mg/kg reduced Ewing sarcoma local relapse and lung metastases, while no effect was observed on the primary tumor growth (Bierbaumer et al., in revision). These results are consistent with our in vitro results demonstrating YAP/TAZ being predominantly involved in tumor cell migration and invasion in Ewing sarcoma. However, it cannot be ruled out that YAP/TAZ-binding independent activities of verteporfin contribute to the observed in vitro and in vivo phenotypes: It reduces both the expression and activating phosphorylation of FAK [151]. It upregulates the chaperon 14-3-3σ, thereby assisting YAP/TAZ sequestration in the cytoplasm [247]. Moreover, it has proteotoxic activity by inducing the formation of cytotoxic high molecular weight complexes, including the autophagosome component p62 (SQSTM1) [248] and the transcription factor STAT3 [249], a target of ROCK2, which was found downregulated in osteosarcoma upon verteporfin treatment [185]. If the ability of verteporfin to induce a cytotoxic oligomer and high molecular weight complex formation is light-dependent or independent remains controversial [243]. 

As YAP/TAZ require tyrosine phosphorylation for transcriptional activity, several clinically approved multi-kinase inhibitors, including dasatinib and pazopanib, were demonstrated to interfere with YAP/TAZ nuclear localization [250] (Figure 4). For example, dasatinib is believed to affect YAP/TAZ primarily through targeting SRC [251]. Studies in both osteosarcoma and Ewing sarcoma cell lines reported inhibition of the migration and induction of apoptosis by dasatinib [252,253]. 

Statins, specific inhibitors of 3-hydroxy-methylglutaryl (HMG) CoA reductase, interfere with the membrane localization of Rho GTPases and actin remodeling and, thus, affect upstream YAP/TAZ signaling (Figure 4). As statins have long been safely used to treat hypercholesterolemia and prevent cardiovascular diseases, they may hold promise as anti-metastatic treatment of cancers, in which YAP/TAZ play an important role [254]. In fact, simvastatin has been demonstrated to exert antimetastatic and cytotoxic effects in osteosarcoma [255,256], at least in part by the downregulation of YAP/TAZ/TEAD target genes CYR61 and CTGF [257,258]. It has also been shown to reactivate YAP-suppressed SOX9 in rat chondrosarcoma cells, leading to chondrocytic re-differentiation [259]. Similarly, lovastatin was reported to reduce the survival of Ewing sarcoma cells [260]. Among epigenetic drugs, only JQ1 was previously reported to affect LATS1, YAP and TAZ in chondrosarcoma [182], potentially by restoring suppressed AMOT expression [183] (Figure 4). Among natural compounds, agave extract was reported to downregulate YAP and TAZ mRNA and the protein in osteosarcoma cell lines by a mechanism that still needs to be explored [261]. Other compounds which can affect YAP indirectly but are not yet tested in bone sarcomas are the synthetic cyclizing-berberine A35 [262] and the traditional Chinese medicinal compound norcantharidin [263]. 

Gene expression databases (like the R2 Genomics Analysis and Visualization Platform (r2.amc.nl)) provide versatile tools to quickly analyze the expression patterns and clinical outcome associations of genes of interest and may be used to interrogate genes involved in YAP/TAZ signaling as potential therapeutic target candidates. Using this approach, we identified an association of high AMOT expression with improved outcomes in both Ewing sarcoma and osteosarcoma (unpublished). Tankyrase, a PARP family member, is known to bind and degrade this negative YAP/TAZ regulatory protein, and tankyrase inhibition was demonstrated to stabilize AMOT and reduce YAP/TAZ nuclear translocation and TEAD target gene expression, potentially offering a therapeutic option in these bone cancers [264,265] (Figure 4). 

YAP and TAZ are often activated as an alternative survival pathway in drug-resistant cells and can be involved in mechanisms inducing drug resistance [27]. One of the most prominent examples is the CDK4/6 inhibitor’s treatment, where YAP and TAZ can upregulate the expression of CDK6 to overcome the pharmacological inhibitory effects [266]. In the context of bone tumors, a recent study suggested that the dual inhibition of CDK4/6 and of IGF-1R may be a candidate synergistic combination for the clinical application in Ewing sarcoma, since CDK4/6 drug resistance is mediated by the activation of IGF-1R signaling [267]. Therefore, the inhibition of YAP and TAZ here may prevent the emergence of drug resistance or restore drug sensitivity [268].

## 10. Conclusions

YAP and TAZ serve key regulatory roles in mesenchymal stem cell development at several steps of osteochondrogenesis. They promote progenitor cell expansion and act as rheostats for terminal osteoblastic differentiation in response to extrinsic and intrinsic biomechanical and biochemical-signaling cues. While functionally redundant at early stages of development, YAP and TAZ functions differ at late stages of osteoblastic differentiation. YAP and TAZ are connected to bone cancer development and metastasis when inappropriately activated in committed osteochondrogenic progenitors. In human bone cancers, YAP and TAZ are variably expressed in the cytoplasm and the nucleus. YAP/TAZ nuclear activation may be assigned to epigenetic silencing or the loss of YAP/TAZ negative regulatory proteins (RASSF family members, NF2, AMOT and MOB1) or to the activation of receptor tyrosine kinases (SRC, FGFR, FAK, etc.) or of chemokine receptor CXCR4 and/or of GPCRs (RHO/RAC). Experimental evidence is compatible with a role of YAP (and, potentially, also TAZ) in tumor stem cell self-renewal, proliferation and migration, while there is accumulating evidence for TAZ being crucially involved in the regulation of tumor cell migration, invasion and metastasis. These findings identify YAP and TAZ as very promising therapeutic targets in cancer in general and particularly for the treatment of bone cancer. However, no directly YAP/TAZ-targeting compounds have been identified so far. In contrast, several small molecules binding to the YAP/TAZ interaction surface of TEAD are currently in development, while broad-spectrum kinase inhibitors targeting YAP/TAZ upstream activating kinases were demonstrated to inhibit YAP/TAZ nuclear activity. 

Despite exponentially growing literature on YAP and TAZ in normal development and disease, there is little discrimination between the tissue/stage and tumor-specific functions of the two paralogues. Moreover, the complex upstream regulatory network and the number of YAP and/or TAZ-interacting regulatory and effector molecules suggest that the mere quantification of cumulative YAP/TAZ expression levels (as is frequently done in translational studies) is unlikely to be of biological significance. Additionally, functional readouts should not be restricted to the monitoring of TEAD activity, as there are also other transcription factors regulated by YAP and/or TAZ. Finally, compounds targeting the YAP/TAZ pathway are expected to shift the subtle balance between self-renewal/proliferation, differentiation, migration and invasion and potentially also apoptosis of tumor cells dependent on tumor type and the biophysical and biochemical properties of the tumor microenvironment. Thus, they affect cellular plasticity but may not suffice to induce widespread cytotoxicity in dormant or disseminated tumor cells. However, YAP/TAZ inhibitors may hold promise as drug sensitizers and should therefore be tested in combination treatments with the standard of care therapeutic drugs. 

## Figures and Tables

**Figure 1 cells-09-00972-f001:**
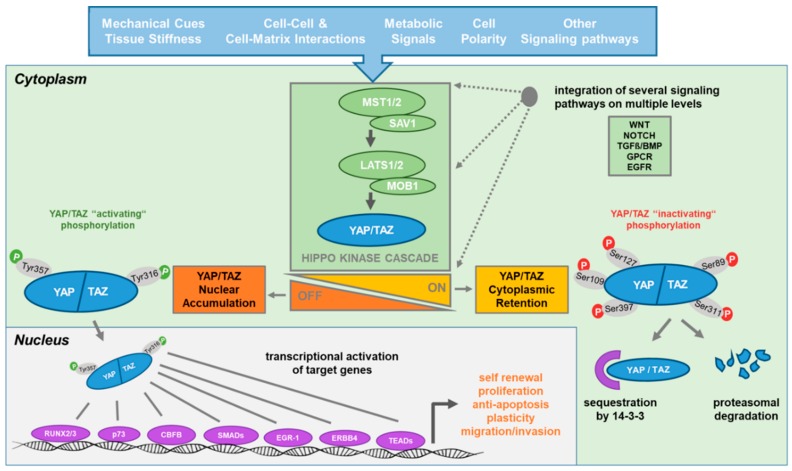
YAP/TAZ activity is tightly regulated—Hippo signaling and beyond. YAP/TAZ subcellular localization and transcriptional co-activator activity in specific contexts are mainly regulated by phosphorylation events. When core Hippo kinases are active (“on”), YAP/TAZ inactivating Ser-phosphorylation events promote cytoplasmic retention and/or degradation, whereas an inactive Hippo kinase cascade (“off”) and Tyr-phosphorylation results in their nuclear accumulation. YAP/TAZ act as prominent links between and integrators of several other signal pathways, such as Notch, GPCR, Wnt, BMP or TGFβ signaling, to name only a few well-studied examples.

**Figure 2 cells-09-00972-f002:**
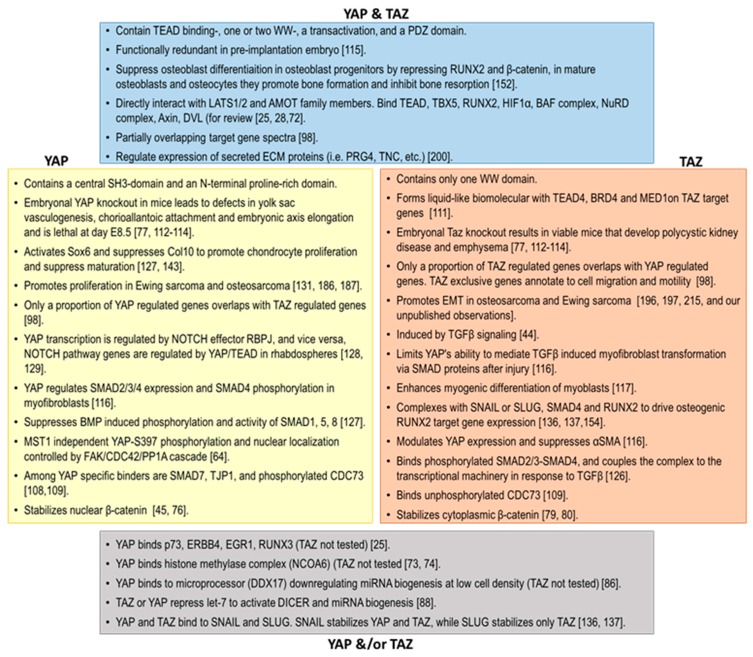
Shared and exclusive biological functions of YAP and TAZ. AMOT, angiomotin; BAF complex, Brahma related gene 1 associated factor complex; BMP, bone morphogenetic protein; BRD4, bromo domain protein 4; CDC42, cell division cycle 42; CDC73, cell division cycle 73; DDX17, DEAD-box helicase 17; DVL, segment polarity protein disheveled homolog; ECM, extracellular matrix; EGR1, early growth response 1; ERBB4, ERB-B2 receptor tyrosine kinase 4; FAK, focal adhesion kinase; HIF1α, hypoxia inducible factor alpha; LATS1/2, large tumor suppressor homologs 1/2; MED1, mediator complex subunit 1; MST1, macrophage stimulating protein 1; NCOA6, nuclear receptor coactivator 6; NuRD complex, nucleosome remodeling and deacetylase complex; p73, tumor protein 73; PP1, protein phosphatase 1; PRG4, proteoglycan 4; RBPJ, recombination signal binding protein for immunoglobulin kappa J region; RUNX2, Runt-related transcription factor 2; SLUG, SNAI family transcriptional repressor 2; αSMA, alpha smooth muscle actin; SNAIL, SNAI family transcriptional repressor 1; Sox6, SRY-box transcription factor 6; TAZ, transcriptional co-activator with PDZ motif; TBX5, T-box transcription factor; TEAD, TEA-domain transcription factor; TGFβ, transforming growth factor beta; TJP1, tight junction protein 1; TNC, tenascin C; YAP, YES-associated protein 1.

**Figure 3 cells-09-00972-f003:**
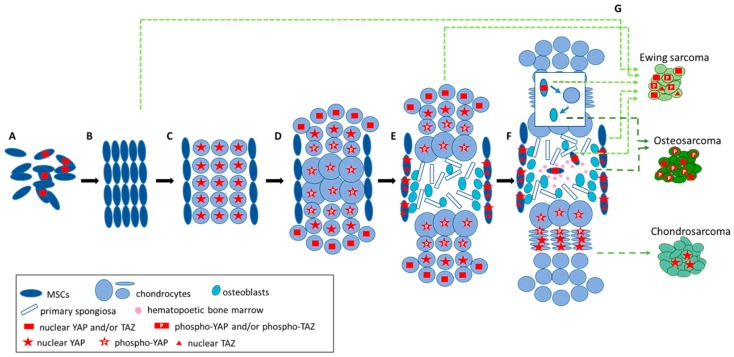
YAP and TAZ in endochondral bone development and most frequent bone cancers. (**A**) Mesoderm and neural crest-derived mesenchymal stem cells (MSC) (dark blue ovals) migrate to the site of limb bud formation. Here, TBX5 plays a pivotal role, for which YAP and/or TAZ (red squares) serve as co-activators. (**B**) MSC condense. Nothing is known about the status and activity of YAP/TAZ at this stage of bone development. (**C**) Condensed MSC differentiate to proliferating chondrocytes (small light blue circles), which produce parathyroid hormone related protein (PTHrP): peripheral MSC form perichondrium. Proliferating chondrocytes express active (nuclear) YAP (red stars). (**D**) Central chondrocytes arrest and become hypertrophic (large blue circles), which express Ser-phosphorylated, active YAP (red stars with inscribed white P) and Indian hedgehog (IHH). (**E**) Hypertrophic chondrocytes generate primary spongiosa (grey rectangles). Stiff extracellular matrix of primary spongiosa promotes the differentiation of perichondrial MSC into osteoblasts (turquois circles). YAP stabilizes nuclear β-catenin, and TAZ (red triangles) binds to SMAD4 co-activating RUNX2 to drive the osteoblastic differentiation program of perichondrial MSC and suppress adipogenic differentiation. These osteoblasts form the bone collar and invade the primary spongiosa to calcify the trabecular bone. (**F**) Growth plate forms columns of proliferating flat chondrocytes (light blue ovals), which express nuclear YAP (red stars), promoting SOX6 and suppressing SOX9 and Col10a1 expression, thus preventing chondrocyte maturation. Hematopoietic bone marrow (pink circles) invades with MSC (dark blue ovals), expressing active YAP and TAZ (red squares). Inserted window: Post-natal osteoblast generation by transdifferentiation from chondrocytes produced by self-renewing PTPrH expressing mesenchymal progenitor cells in an epiphyseal stem cell niche. Here, YAP and/or TAZ likely play a role in migration to sites of injury and bone regeneration. (**G**) Most frequent bone cancers with their hypothesized alternative origins indicated by dashed arrows. Ewing sarcoma most likely deriving from either early mesoderm or neural crest derived embryonal MSC, osteochondrogenic progenitors of the superficial zone or of the epiphyseal stem cell niche or from bone marrow MSC. Ewing sarcoma expresses nuclear and cytoplasmic YAP and TAZ (red squares with/without inscribed white P), but TAZ and TEAD-target gene activation is upregulated in rare tumor cells supposed to express low levels of EWS-FLI1. Osteosarcoma most likely deriving from osteoblasts expresses high levels of mostly cytoplasmic YAP (red stars with inscribed white P) with nuclear YAP (red stars), being associated with a bad prognosis. Chondrosarcoma supposed to originate from growth plate chondrocytes shows nuclear YAP (red stars) accumulation as a consequence of epigenetic large tumor suppressor kinases (LATS) inactivation as an independent bad prognostic factor. For simplicity, the figure does not consider the important role of osteoclasts in bone turnover.

**Figure 4 cells-09-00972-f004:**
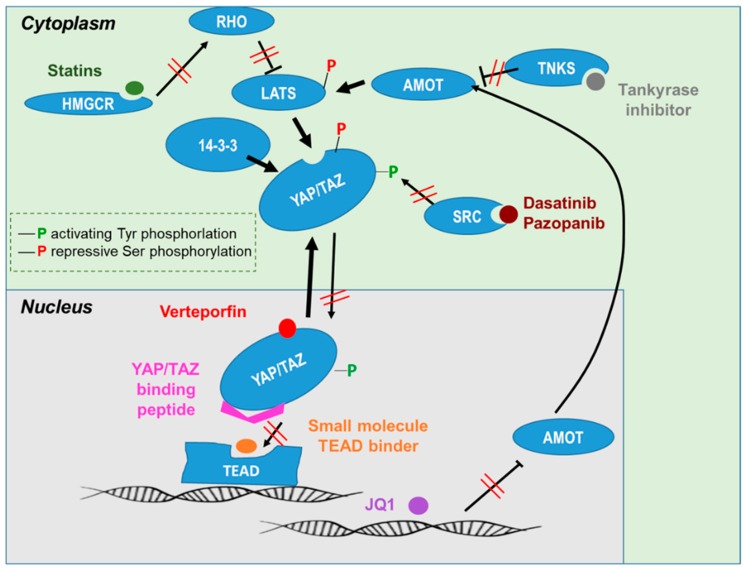
Therapeutic options to inhibit YAP/TAZ activity. HMGCR, β-hydroxy-β-methylglutaryl coenzyme A reductase; RHO, RAS homolog family member; SRC, non-receptor protein tyrosine kinase SRC; TNKS, tankyrase.

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
