# Peer review of "The YAP/TAZ Pathway in Osteogenesis and Bone Sarcoma Pathogenesis"

_cells, 2020, doi:10.3390/cells9040972_

Round 1

Reviewer 1 Report

This excellent review comprehensively covers all important aspects of normal bone development and bone cancer in which TAZ / YAP are involved. Literature compilation is impresive and the manuscript is well structured. Only a few comments/suggestions need to be addressed:

  • It should be made clear in the introduction that TAZ and YAP are transcriptional co-activators that are active in the nucleus. Both the concept of nuclear translocation and that of transcriptional regulators that do not directly bind DNA are extensively referered afterwards in the whole manuscript. Likewise, it may be worth to state in the introduction that TAZ and YAP are the final effectors of Hippo developmental pathway, so that the reader is able to better follow the concepts presented in section 2.1. Indeed, a figure for section 2.1 depicting the core components of the Hippo pathway would be very helpful to those unfamiliar with it.

  • Regarding the role of TAZ/YAP in miRNA biogenesis, it would be interesting to discuss how this is reconciled with the general downregulation phenotype of miRNAs in cancer.

  • The authors discuss that surrogate markers frequently used to evaluate TAZ/YAP activity are not accurate since TAZ/YAP are not functionally redundant and not restricted to TEAD activation. It would be interesting to bring forward some alternatives to address this issue.

  • Fig. 1 could be converted to a table adding information about the reports accounting for each statement.

  • The authors should discuss on the possible role of TAZ/YAP in coupling EMT phenotype with stemness, as well as the connection with scribble, which organizes basolateral cell polarity.

  • It should be specified whether phosphorylations of TAZ/YAP are activating or inactivating in Fig. 2 and 3 and other parts of the text.

  • Alternative promoter activation upon RASSF1 locus hypermethylation leading to RASSF1C expression should be better explained.

  • TAZ and YAP are described as orthologous genes in several part of the text, whereas they are referred to as paralogs in the introduction. In fact, TAZ is belived to be originated from gene duplication of YAP in a vertebrate ancestor. The same occurs with other members of the core of hippo signaling, which confers increased functional complexity and versatility to the pathway. Therefore, the correct designation is paralogs, since orthologous refers to homologous genes that diverged in different species (i.e. YAP, Yki ortholog).

  • It is clear that post-translational modifications are the main mechanisms modulating TAZ/YAP activity. However, it would be interesting to know additional information or studies addressing the transcriptional regulation of TAZ/YAP. Are they constitutively transcribed or rather induced in particular contexts?

Reviewer 2 Report

The manuscript is a review entitled "the YAP/TAZ pathway in osteogenesis and bone sarcoma pathogenesis" by Kovar et al.

This review is a dense (maybe too dense) review on the current knowledge about YAP/TAZ signaling in bone and the related disease: bone sarcoma. It is well written, and the overall structure intelligible.

However I would recommend couple of changes to make this review more comprehensive for a broad readership.

Main points :

In the first part, in which the authors discuss signaling pathway :

  • Addition of figures to support the text would be useful.
  • 1 paragraph would require some changes to make the authors point more evident. There is a lot of information but the reader is left without a clear picture.
  • Line 81 the word ‘integrate’ is not appropriate, YAP and TAZ transmit signals from the plasma membrane or metabolic cues to the nucleus.
  • Line 98 “14-3-3σ” is restrictive, to the best of my knowledge the binding of YAP/TAZ is not restricted to this protein, but rather shared by other 14-3-3 members.
  • Latter in this paragraph, the authors discuss the role of FAK and downstream signaling. Here there is a bench of date from literature that are missing that would be important to incorporate. Indeed, it starts to become clear integrin integrate extracellular matrix cues by activating a pathway in which the Src non receptor tyrosine kinase has a major role (Lamar et al., Sabra et al. etc…). A more complete description of this pathway is important, as Src is emerging as a potential therapeutic targets in oncology.
  • Paragraph 2.2 and 2.3. I don’t really understand why the authors discuss these issues. Is that relevant for bone or bone sarcoma. Please either remove or make the link more obvious.
  • 4 : YAP and TAZ are not sensor they are relays of mechanical stimulation. Sensor are at the plasma membrane. They transmit those informations from the integrin and likely cadherins to the nucleus.
  • 5. This is an important issue that is poorly addressed in many other reviews. The authors should consider to incorporate and discuss the recent work from Lu et al. NCB, 2020.

In the part YAP and TAZ in osteogenesis.

  • Here my main concerns is the poor consideration of the osteoblastic population. While a short introduction is presented on bone formation, very few to discuss the relative contribution of different bone cells function (osteo-formation vs osteo-resorbtion).

Minor point :

  • I am wondering why the authors left behind important emerging findings such as metabolic state and autophagic process in YAP/TAZ biology. This could have been relevant in particular when tumor is on the table.

Reviewer 3 Report

In this review, Kovar et al describes the role of YAP/TAZ in conferring an invasive, metastatic and EMT phenotype to cells. They briefly discuss the signaling events that regulate YAP/TAZ, therapeutic opportunities and more importantly they elaborate the functions of YAP/TAZ in bone formation and how their activity is dysregulated in bone cancers, which is the focus of this review. This is a niche area and there aren’t many reviews, that is why I recommend publication of this review in “Cells”. In particular, I like Figure 2, where the authors meticulously described the role of YAP/TAZ at each stage of endochondral bone development. However, the following are my recommendations to improve the review:

1.In many instances, I have noticed that the authors cite reviews instead of primary research.

  1. Introduction - for stem cell renewal, Varelas X et al, Nat Cell Biol, 2008 primary research should be cited.
  2. Introduction – which papers define a role for AREG and CYR61 in modulating tumor microenvironment and in promoting metastasis.
  3. Rare cancers – TAZ-CAMTA1 fusion – Tanas M et al Sci Transl Med 2011 & Errani, C et al, Genes Chromosomes Cancer, 2011

, to name a few.

  1. To make it complete, in addition to mentioning integrins in section 2 (key elements of YAP/TAZ signaling), E-cadherin and Crumbs could also be mentioned as they also sense mechanical and architectural cues.

  1. Line 102, authors talk about Mst/Hippo independent regulation of LATS by KIBRA but ref 41 cited by authors indicates otherwise i.e. Hippo-dependence.

4. Other relevant works to include:

a.AMPK regulation of YAP should also be mentioned in section 2.1.

b.Targetability of TEADs/palmitate-binding pocket (Pobbati et al, Structure, 2015) under 9. Therapeutic opportunities.

  1. In section 2, it is misleading to emphasize YAP/TAZ cross talk with only Wnt signaling why Wnt deserved a special mention (2.2) while other signaling pathway cross talks were ignored.

6.Some typos that I had noticed

a.Frizzeled, line 126

b.Hyperthrophic, line 339
